# Incorporating the Cycle Inductive Bias in Masked Autoencoders

Start Gallina Ottersen*[1] and Kerstin Bach[1]

[1]Norwegian University of Science and Technology, Computer Science department

## Abstract

Many time series exhibit cyclic structure — for example, in physiological signals such as ECG or EEG — yet most representation learning methods treat them as generic sequences. We propose a masked autoencoder (MAE) framework that explicitly leverages cycles as an inductive bias for more efficient and effective time-series modelling. Our method decomposes sequences into cycles and trains the model to reconstruct masked segments at both the cycle and sequence level. This cycle-based decomposition shortens the effective sequence length processed by the encoder by up to a factor of ten in our experiments, yielding substantial computational savings without loss in reconstruction quality. At the same time, the approach exposes the encoder to a greater diversity of temporal patterns, as each cycle forms an additional training instance, which enhances the ability to capture subtle intra-cycle variations. Empirically, our framework outperforms three competitive baselines across four cyclic datasets, while also reducing training time on larger datasets.

## 1 Introduction

Sequential data is ubiquitous, as many real-world phenomena occur sequentially in time. Prominent examples include health, finance, climate, and speech, making the ability to model such data essential. As a subset of these data, we have cyclic data commonly found in biological signals such as Electrocardiogram (ECG, recording of the heart's electrical activity), Electroencephalogram (EEG, recording of the brain's electrical activity), or respiration, environmental data (seasons, tides), and industrial processes that often exhibit regular rhythms. An example of the cyclic pattern of an ECG can be seen in Figure 1. Modelling cyclic structures and their abnormalities is crucial for tasks such as arrhythmia detection in ECG, identifying irregular brain rhythms in EEG, and monitoring disordered sleep cycles.

The cyclic structure can support modelling the overall sequences, but has some inherent challenges where the periodicity can interact with noise, as well as general inter-individual variability, which can make modelling harder than purely sequential settings. This presents an opportunity for self-supervised learning like Masked Autoencoders (MAE), which can leverage large unlabeled datasets, reducing dependence on expensive expert labelling.

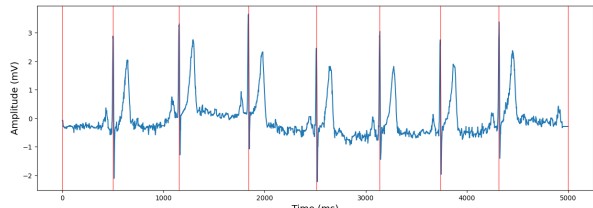

**Figure 1.** Example of ECG including cycle segmentation.

In self-supervised learning, the model learns from automatically generated tasks derived from the data, such as reconstructing missing or masked portions. This has been shown to be highly effective in language modelling and more recently on image data, and now on time series as well, used by, among others, the MAE [1, 2]. The MAE is based on an assumption that visible areas of the data contain information about the masked areas and that these masked areas can be reconstructed by encoding the visible ones in an effective way. This assumption is called an *inductive bias*. The encoding is often performed using a transformer, and although highly effective, the transformer is also computationally very expensive, making it challenging to apply directly on long time series, which is becoming more common with the lower cost of high-frequency sensors. To mitigate the increase in computational requirements, other works decompose the time series into patches, reducing the quadratic complexity with respect to the input sequence length [3–5]. Analogous to how NLP models segment text into tokens, this approach can be seen as moving toward a more semantically grounded partitioning of the time series.

In this paper, we suggest an MAE using cycles as an inductive bias and how this can be used as a way to reduce computation in longer sequences while maintaining performance. The underlying idea is to apply the transformer encoder at two stages, first for each cycle and second for a sequence of class tokens that each represent a cycle. This allows for more efficient use of the encoder by reducing each sequence to individual cycles, decreasing the sequence length by up to orders of magnitude, while

---

*Corresponding Author.

Proceedings of the *7th Northern Lights Deep Learning Conference (NLDL)*, PMLR 307, 2026.

also increasing the number of individual time series the encoder is exposed to.

## 2 Related Work

Originating in language pedagogy, the cloze procedure, which is similar to masked learning, was a measure to test readability with every n-th word removed [6]. This was later transferred to NLP in Baevski et al. [7] and further popularized in BERT [1], where it became the masked learning we know today. The use of masking as a self-supervised task was introduced to non-text learning in "Masked Autoencoders Are Scalable Vision Learners" [8]. It is only more recently that this same approach is being applied to time series with papers such as TimeMAE, ExtraMAE, Cross Reconstruction Transformer (CRT), Masked Autopencoder for Multivariate Time-Series (MTSMAE), MOMENT [2, 3, 5, 9, 10]. These all base themselves on the structure popularised with the MAE; the original structure was designed for image data rather than time series. We therefore see representational or structural changes to adapt the model. For example, CRT using Fast Fourier Transforms to include phase data during pre-training [8]. Another example is MTSMAE doing causal masking to avoid information from future steps being included when predicting past data to improve forecasting performance [9]. ExtraMAE, on the other hand, uses multiple partial observations so that each portion of a time series is at one time masked, and these reconstructions are merged to form a complete synthetic time series reconstruction [10]. TimeMae takes a different approach and adds codebook representations, allowing it to indirectly reconstruct the signal by turning it into the classification problem of selecting which codebook token fits best in a given masked patch [3]. The MAE has now also been used in a foundational model for time series with MOMENT, which is trained on a wide variety of time series data and shows exceptional performance on a wide variety of tasks [4]. On classification tasks, the performance is state-of-the-art, and the model performs well on unseen datasets without finetuning as well. One of the main advantages of foundational models is the ability to do finetuning at a lower cost and with better performance than models trained only on the task at hand.

## 3 MACE: Masked Autoencoder Cyclic Encoder

In this section, we introduce Masked Autoencoder Cyclic Encoder (MACE), which is a masked autoencoder-based model that processes cyclical sequential data as separate cycles as part of the masked reconstruction task it performs during pre-training.

The model leverages a siamese network for the transformer encoders, which is used to regularise and give a rich representation with full attention as a reconstruction task for the student encoder. The main cost of these models is the quadratic scaling of self-attention to the sequence length, which can quickly become prohibitively expensive. Alternative architectures such as the LinFormer and MAMBA exist, but instead of focusing on computationally intensive architectures, we aim to demonstrate that incorporating the inductive bias of cyclic data can yield competitive results while requiring far fewer resources [11, 12].

Using cycles as sub-components of sequences, beyond efficiency, also exposes the encoder to data with more consistent structure, since cycles must align closely to be treated as comparable. This approach produces representations that capture fine-grained variations in cycles — an ability that is essential in tasks like arrhythmia detection and broader cardiovascular diagnostics.

### 3.1 Pre-training structure

The pre-training structure can be seen in Figure 3. MACE receives a sequence $S$ with one or more channels and a set of cycle start indices $0, n_1, n_2, ...n_k, T$ where T is the length of the sequence $S_i$ and k is the number of cycles in the sequence. $S_i$ is then split into a series of cycles $c_{i0}, c_{i1}, ..., c_{i(k-1)}, c_{ik}$. Due to the potential varying length of the cycles, each cycle is padded to be the same length. Each cycle $c_j$ is subsequently split into patches $p_{ij0}, p_{ij1}, ..., p_{ij(k-1)}, p_{ijk}$ of equal length m, this is again to reduce computational cost and due to the large similarity between adjacent data points. Each patch is embedded with a linear layer, and we then add an absolute positional embedding [13]. We then prepend a class token $q_ij$ to each cycle $c_j$ so the cycle can be represented by $q_{ij}, p_{ij0}, p_{ij1}, ..., p_{ij(k-1)}, p_{ijk}$ [1]. A random mask is then applied to each cycle $c_j$. The mask is designed so that 75% of the sequence $S_i$ is masked, and the mask is then split into masks for the cycles. The masking is shown in Figure 2, where one can see that the masking percentage of the cycles can vary while the masking percentage of the sequence remains constant.

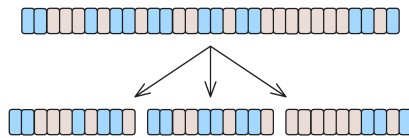

**Figure 2.** Masking of sequences and cycles

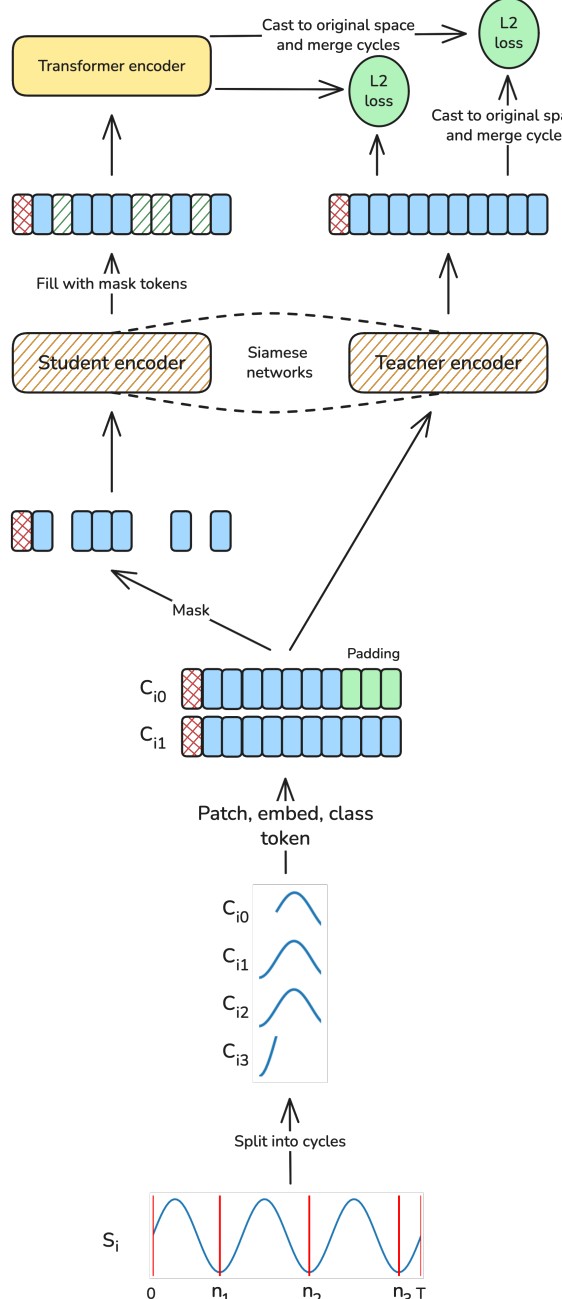

**Figure 3.** Pre-training architecture of the proposed Masked Autoencoder Cyclic Encoder (MACE): Sequential data is decomposed into cycles for masked reconstruction, with a siamese transformer encoder setup providing regularisation and efficient representation learning.

The same cycles are passed to the teacher encoder, including the class tokens, without any masking performed. The teacher model produces an encoded cycle with full attention that can be used as a reconstruction target for the student model. The teacher encoder does not learn through backpropagation, but is rather updated as a moving average of the student encoder [14]. Thereby, it creates a rich representation with full attention over the entire sequences that can be used as the target for the main encoder. After passing the unmasked areas through the student encoder, the masked areas are filled with a learnable mask token. This reconstruction is passed through a lightweight decoder, which is a transformer encoder.

We then calculate two losses, one for the cycles by directly comparing the output of the teacher encoder with the cycle reconstruction. The second loss is calculated by creating two full sequence reconstructions, one using the teacher encoder output and another using the reconstructed cycles. The full sequence reconstruction is created by merging the cycles into sequences and casting it back into the original space with a linear layer. Using these two losses, we get a rich representation of the cycles in the embedding space while also evaluating the full reconstruction of the sequence in the original space. The loss is only calculated using the masked areas of the data and is calculated as follows:

$$\mathcal{L}_{\text{cycle}} = \frac{1}{|M|} \sum_{i \in M} (\tilde{p}_i - \hat{p}_i)^2$$

$$\mathcal{L}_{\text{sequence}} = \frac{1}{|M|} \sum_{i \in M} (\tilde{d}_i - \hat{d}_i)^2$$

$$\mathcal{L} = \mathcal{L}_{\text{sequence}} + \mathcal{L}_{\text{cycle}}$$

Where $\tilde{p}_i$ is a patch in the teacher encoder representation of the sequence and $\hat{p}_i$ is a patch in the cycle reconstruction, these are both in the embedding space. $\tilde{d}_i$ is a data point in the full sequence representation, and $\hat{d}$ is a data point in the sequence reconstruction; these are in the original space of the data. $|M|$ represents the mask, so only the masked patches/data points are included in the loss.

## 3.2 Fine-Tuning Structure

The fine-tuning structure of MACE is presented in Figure 4 and shows how the encoder is reused from the pre-training to create rich representations of the data. All of the previously discussed models directly use the embeddings from the encoder in a linear layer to classify [2, 3, 8–10]. MACE, on the other hand, uses a multi-head attention to query the class tokens from a sequence rather than simply using a linear layer. A sequence contains multiple cycles $c_{i0}, c_{i1}, ..., c_{i(k-1)}, c_{ik}$ and each cycle has its class token $q_i j$, we keep only the class tokens and merge them back into sequences. Meaning a sequence $S_i$ is represented only by class tokens $q_{i0}, q_{i1}, ..., q_{i(k-1)}, q_{ik}$. This representation of the sequence is then queried by a trainable "query token" using multi-head attention. The attention information is then used to classify using a linear layer.

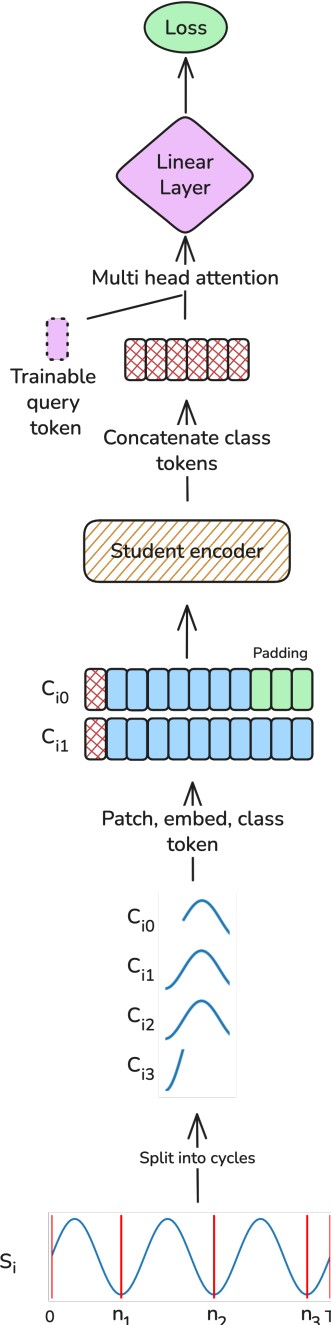

**Figure 4.** Fine-Tuning structure of MACE: the pretrained encoder generates class tokens for each cycle in a sequence, which are merged and queried by a trainable token through multi-head attention. The resulting representation is then classified via a linear layer.

## 4 Experiments and Results

We conduct experiments on six different datasets, four of which contain cycles: UCI Electricity Load Diagrams, UCI PM2.5 Data of Five Chinese Cities (Air quality), NHANES activity 2013-2014, and PT-BXL. These were selected due to their varied sampling frequency, sequence length, and size of datasets [15–18]. *HAR* and *Ford A* were also included to test

the model's capabilities on semi-cyclic data [19, 20]. Table 1 provides an overview of the characteristics of each dataset.

### 4.1 Data

**PTBXL** The PTB-XL dataset is a large publicly available collection of electrocardiogram (ECG) signals, containing clinical 12-lead recordings of 10 seconds each at 500Hz [18]. There are six classes: Normal ECG, Myocardial Infarction, ST/T Change, Conduction Disturbance, and Hypertrophy. One instance can be part of multiple classes, making it a multilabel problem where we try to predict all labels. For this dataset, cycles are detected using the neurokit library to detect heartbeats [21].

**UCI Electricity Load Diagrams (ELD)** The UCI Electricity Load Diagrams dataset includes hourly electricity consumption measurements from 370 households and businesses in Portugal over a four-year period (2011–2014) [16]. In our work, we split the data into week-long segments for each household and gave a label from 0 to 3 based on what quantile of total consumption the household is in for the whole year. The data is split strictly on household to avoid data leakage. So the task is to classify each week, predicting what quantile the given household will be in. The data contains natural cycles of equal length, so the week-long segments are split into days from midnight to midnight.

**Air quality** Hourly data from five major Chinese cities on air quality and meteorological data. We only use the features on PM2.5 gathered at the US embassies, dew point, temperature, humidity, and pressure[17]. The data is segmented into week-long sequences and given a label based on the city where it was measured. Some missing values are present, but are ignored if the sequence contains at least 160 measurement per week, meaning a sequence can miss maximum 8 hours of measurements per week, otherwise the sequence is discarded. The task is to predict which city the week-long measurement is coming from. The data contains natural cycles of equal length, so the week-long segments are split into days from midnight to midnight.

**NHANES activity 2013-2014** The NHANES activity dataset contains accelerometer-based actigraphy recordings collected as part of the National Health and Nutrition Examination Survey [15]. The data is collected with a wrist-worn actigraph for 7 days at 80Hz; we only use the minute-averaged data that is publicly available and further average it into 30-minute segments. The classification task is to categorise each segment into 4 age quantiles. The data contains

natural cycles of equal length, so the week-long segments are split into days from midnight to midnight.

**UCI HAR** The UCI Human Activity Recognition (HAR) dataset contains smartphone sensor recordings from 30 participants, including accelerometer and gyroscope measurements at 50Hz for 2.56 seconds [19]. The participants completed 6 different activities: walking, walking up stairs, walking down stairs, sitting, standing, and lying, which are used as the labels for the classification task. We use the suggested train test split and create a validation set by randomly selecting 20% of the train set's participants. To detect peaks, we use a Savitzky-Golay filter for smoothing and scipy's built-in *find peaks* function [22].

**FordA** The FordA dataset, part of the UCR Time Series Classification Archive, consists of univariate time series of length 500 collected from Ford automotive engine sensors [20]. It is a binary classification problem with normal versus abnormal engine condition labels. To detect peaks, we use a Savitzky-Golay filter for smoothing and scipy's built-in *find peaks* function [22].

| Dataset | Classes | Channels | Instances | Sequence length | Patch size |
|---|---|---|---|---|---|
| PTB-XL | 5 | 12 | 21799 | 5000 | 20 |
| UCI ELD | 4 | 1 | 75110 | 672 | 7 |
| Air quality | 5 | 5 | 750 | 160 | 4 |
| NHANES Age | 4 | 1 | 7511 | 384 | 4 |
| UCI HAR | 6 | 9 | 10299 | 128 | 8 |
| FordA | 2 | 1 | 4921 | 500 | 5 |

**Table 1.** Characteristics of the datasets used in the experiments: number of classes, number of channels, number of instances, length of the sequence per instance and patch size used during experiments.

## 4.2 Experimental setup

The models are pretrained on the entire dataset before they are finetuned on the labels. All the models are fully finetuned except MOMENT due to computational cost limitations. Fully fine-tuned, meaning all trainable parameters are updated during training, rather than, for example, linear probing. MOMENT is evaluated similarly to its original paper, where two separate results are presented [4]. We run a 5-fold cross-validation on all datasets and stop after

100 epochs. During fine-tuning, an early stop with a patience of 10 is used. We use an embedding size of 128 and a 4-layer encoder with 4 heads.

## 4.3 Results and Discussion

Table 2 and Table 6 show the results of the experiments.

MACE is able to outperform the three baseline models on all four cyclic datasets (shown in Table 2). These datasets contain a wide range of frequencies, timescales, and contexts and should give a broad test of the model's capabilities. The running time for MOMENTs linear probing was too expensive, and results for it were not gathered; the running times are shown in Table 3. Interestingly, we see that *MOMENT* struggles on the cyclic datasets, falling short by a considerable margin, which is likely due to the type of data presented being outside of the distribution of datasets the model is pretrained on. On *UCI ELD*, *MACE* and *TimeMAE* outperform the other models by a large margin. The better performance can likely be attributed to their common teacher-student structure with dual encoders, which helps regularise. Regularisation is particularly important for the *UCI ELD* dataset, as the training and test sets are drawn from different distributions, meaning that the households contributing electricity load data differ between the two sets. *MACE* performs considerably better than the other models on *Air quality*, which is likely due in part to the regularisation of the teacher-student structure, but also the local structure importance in the dataset. The dataset models weather and air quality over a week, and the model is trying to predict which city the measurement is from. The long term dependencies become less important in differentiating the cities as the behaviour is likely similar, its rather the subtle differences in local cycles that help in differentiating the climate in the different cities.

Table 3 shows the running times for the fine-tuning of the various models on the *PTB-XL* and *Air quality* datasets. We use a dedicated V100 GPU for each run. The datasets were chosen as they are the largest and smallest datasets included in the testing. In Table 3, we see that the fine-tuning time for *MACE* is considerably lower than the

| | MACE | MOMENT SVM | MOMENT lin. probe | CRT | TimeMAE |
|---|---|---|---|---|---|
| PTB-XL | **70.50±0.56** | 46.18±0.29 | - | 67.64±1.72 | 64.55±2.44 |
| UCI ELD | **78.17±3.23** | 41.01±2.39 | 41.34±2.00 | 29.90±14.86 | 76.06±2.77 |
| NHANES Age | **63.42±1.04** | 56.38±1.29 | 55.90±1.52 | 61.87±1.72 | 58.25±1.45 |
| Air quality | **85.64±2.39** | 60.49±3.75 | 40.21±3.60 | 48.95± 3.08 | 70.63±3.92 |

**Table 2.** F1 scores for cyclic data, macro averaging is used in multiclass problems.

other models on *PTB-XL*. *MOMENTs* linear probing fine-tuning time also becomes prohibitively expensive on *PTB-XL*. *PTB-XL* is especially problematic for *MOMENT* as it has a long sequence and multiple channels, which becomes a problem as *MOMENT* treats each channel as a separate univariate sequence. Full fine-tuning was not attempted on *MOMENT* as it requires considerably more computation than a simple linear probing. On the much smaller *Air quality* dataset, we see that the pretrained *MOMENT* with an SVM and TimeMAE are considerably faster. Indicating that there is an overhead in *MACE* that scales efficiently, but does not perform well on smaller datasets. This overhead is mostly due to reshaping and padding, which are not very optimised on GPUs, as well as the use of multiple transformer encoders. The running times are just one single run due to computational limitations, as dedicated GPUs are required to make the experiments comparable; the values should nonetheless give an idea of relative running times.

| Model | Running time PTB-XL | Running time Air quality |
|---|---|---|
| MACE | 44m 54s ± 5m 38s | 4m 3s ± 1m |
| MOMENT SVM | 4h 56m 53s ± 29m 52s | 15s ± 3s |
| MOMENT lin. probe* | 400h | 9m 24s ± 2m 29s |
| CRT | 2h 6m 56s ± 3m 10s | 3m 44s ± 16s |
| TimeMAE | 1h 9m 33s ± 2m 2s | 26s ± 10s |

**Table 3.** Fine-tuning times. *Linear probing time is estimated from running time after 12 hours.

We further try to understand what makes *MACE* fine-tuning time shorter than the other models by comparing the losses. The graph in Figure 5 is only produced for the fine-tuning, as both *TimeMAE* and *MACE* use teacher-student, giving the loss a "moving target", meaning the loss can increase while the model is still learning. The losses are presented in Figure 5 where we see that *MACE* converges considerably quicker to its best loss when compared with *TimeMAE* and *CRT*, explaining in part why the fine-tuning time is considerably shorter. The graphs are cut off before the 100-epoch mark due to early stopping. For *CRT*, we see a steep loss at epoch 50 as the model starts full fine-tuning instead of only linear probing.

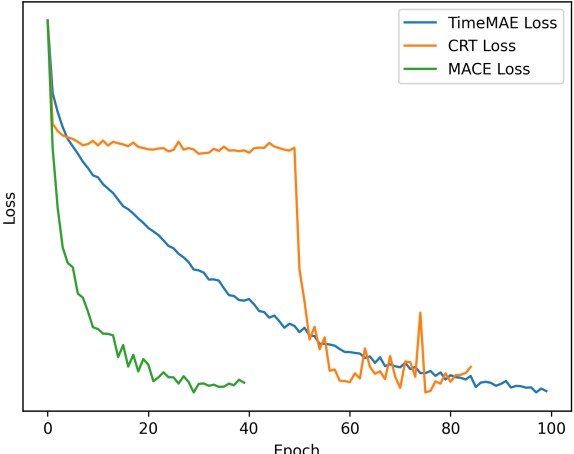

**Figure 5.** Loss curves during fine-tuning of the TimeMAE, CRT and MACE models. The graphs are cut off due to the early stop criterion.

MACE employs two losses, and to test the effect of each of these, we performed pretraining using each loss individually and in combination. The results can be seen in Table 4, where one can see that the results overall are relatively similar, where almost all the results of only using cycle or sequence loss are within the standard deviation of the results using both the losses. The exception is the air quality dataset where the combination of the two seems to make a considerable difference in the performance. This is also supported by the larger discrepancy between only cycle loss and only sequence loss in the Air quality dataset compared to the other datasets.

| | Cycle | Sequence | Cycle+ Sequence |
|---|---|---|---|
| PTB-XL | **70.81±0.71** | 70.67±0.66 | 70.50±0.56 |
| UCI ELD | **79.25±2.99** | 78.11±3.72 | 78.17±3.23 |
| NHANES Age | 64.23±2.06 | **64.72±1.36** | 63.42±1.04 |
| Air quality | 79.85±4.00 | 82.37±3.05 | **85.64±2.39** |

**Table 4.** F1 scores for MACE pre-trained using different losses.

We also conduct an ablation study to test the effect of using a teacher encoder, comparing it to evaluating the reconstruction on the raw input data (Table 8). We observe slight differences in performance, with a slight advantage to not using a teacher encoder, except for the *Air quality* dataset, where the teacher encoder significantly improves performance and NHANES Age where no teacher encoder performs better.

| | No teacher encoder | Teacher encoder |
|---|---|---|
| PTB-XL | **71.53±0.28** | 70.50±0.56 |
| UCI ELD | **78.73±3.62** | 78.17±3.23 |
| NHANES Age | **66.18±1.36** | 63.42±1.04 |
| Air quality | 78.77±3.15 | **85.64±2.39** |

**Table 5.** F1 scores for MACE pre-trained using different learning representations.

|         | MACE | MOMENT SVM | MOMENT lin. probe | CRT | TimeMAE |
|---------|------|------------|-------------------|-----|---------|
| UCI HAR | 90.96±0.54 | 73.85±0.97 | 71.30±0.85 | 92.34±0.44 | **93.80±0.80** |
| FordA | 88.20±1.59 | 90.42±0.54 | 87.51±0.79 | **94.86±3.20** | 87.86±2.67 |

**Table 6.** F1 scores for semi-cyclic data, macro averaging is used in multiclass problems.

We also test the effect of varying masking ratios during training in Table 7, showing slight improvements using 25% masking as the best, with the exception of *Air quality*, where 75% performs the best.

|            | 0.25 | 0.5 | 0.75 | 0.9 |
|------------|------|-----|------|-----|
| PTB-XL     | **70.90±0.82** | 70.79±0.52 | 70.50±0.56 | 70.49±0.51 |
| UCI ELD    | **79.34±2.60** | 79.28±2.39 | 78.17±3.23 | 78.64±3.85 |
| NHANES Age | **65.07±1.10** | 64.76±1.04 | 63.42±1.04 | 64.76±1.31 |
| Air quality | 83.14±2.84 | 82.79±2.95 | **85.64±2.39** | 80.64±2.17 |

**Table 7.** F1 scores for MACE pre-trained using varying masking rates.

In addition, we tested the effect of query attention in the fine-tuning step and compared it to mean pooling, a simpler alternative. Mean pooling appears to outperform query attention, but all values are within a standard deviation; the variation using mean pooling also seems to increase.

|            | Mean pooling | Query attention |
|------------|--------------|-----------------|
| PTB-XL     | **71.04±0.80** | 70.50±0.56 |
| UCI ELD    | **78.85±3.25** | 78.17±3.23 |
| NHANES Age | **64.24±1.35** | 63.42±1.04 |
| Air quality | **86.60±3.33** | 85.64±2.39 |

**Table 8.** F1 scores for MACE pre-trained using different learning representations.

Further, to explore the generalisability of *MACE*, we test it on datasets that fall slightly outside the inductive bias of the model. Meaning datasets that are semi-cyclic. We therefore include *UCI HAR* and *Ford A*. *UCI HAR* is an activity classification problem with actigraphy from a smartphone, where three of the classes: walking, walking up stairs, and walking down stairs are cyclic, while sitting, standing, and lying do not have any cycles. *Ford A* is a fault detection task with sensors on running engines. The data does have cycles, but they vary in length and are noisy. As the data is collected from engines running, there is likely more than one cyclic process registering at once, making it difficult to detect and process each segment as a meaningful and equivalent cycle.

In Table 6, we see that *MACE* is able to out-compete some models, but falls short of the best performing ones on both datasets. On *UCI HAR*, we see *MACE* struggling with the non-cyclic classes; this is likely due to the whole sequence being passed through the model as if it were a single cycle. On *Ford A*, the performance is comparable to the other models except *CRT*; the *CRT* model has a considerable performance boost due to the use of phase data from the Fast Fourier Transform. Running *CRT*

without phase information reduces the performance to 61.96±4.75.

## 5    Conclusion

In this paper, we present a novel variation of the masked autoencoder with an inductive bias for cyclic data, allowing for improved performance at a lower computational cost in longer sequences. The approach is compared to three baseline methods on four cyclic datasets and has better performance on all of them. The model additionally shows comparable performance on semi-cyclic data, which falls outside of the scope of the model. The running time of the presented model shows great performance compared to the other models on large sequences with multiple channels, but has some overhead, which makes the performance worse on smaller datasets. The decreased running times on large sequences and good classification performance overall indicate that the cyclic inductive bias is useful to the model and aids in producing good representations while holding fine-tuning costs down. Future work should explore in more detail how the masking affects learning on time series, as this may lead to more generalisable approaches compared to the method explored in this paper.

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
