# OpenReview forum: "Incorporating the Cycle Inductive Bias in Masked Autoencoders"
_NLDL.org/2026/Conference — NLDL 2026 Spotlight_

### Official Review · Reviewer_5G22 · 2025-09-22
**Review 1**

**Rating:** 4
**Confidence:** 4

**Summary:**

The paper proposes MACE, which first splits a time series into cycles, patches each cycle, masks most content, and trains a student encoder to reconstruct masked parts while a teacher encoder (an exponential moving average of the student) provides stable targets from unmasked data. For fine-tuning, only the per-cycle class tokens are kept; a learnable query token attends over these to form the final representation for classification.

Empirically, MACE beats TimeMAE, CRT, and MOMENT across four cyclic datasets (PTB-XL, UCI ELD, Air Quality, NHANES Age) and remains competitive on two semi-cyclic datasets (UCI HAR, FordA), though it lags on non-cyclic classes and settings where phase features favor other methods. Reported fine-tuning time is substantially lower on large, long sequences, supporting the practical efficiency claim.

**Strengths:**

Clear and well-motivated inductive bias that matches cyclic domains such as ECG and energy load.

Shorter effective inputs and a light decoder lead to faster fine-tuning without obvious loss in quality.

Consistent gains in macro-F1 on diverse cyclic datasets, suggesting better representations rather than just speedups.

Sensible design choices: teacher–student regularization, two reconstruction levels, and a lightweight query-over-tokens aggregator.

**Weaknesses:**

Heavy reliance on accurate cycle detection, with no robustness analysis for misaligned or irregular cycles; this is critical for practical use.

Efficiency evaluation focuses on wall-clock time; missing FLOPs, memory usage, and accuracy-versus-cost trade-offs.

Baselines do not include several strong recent time-series models or efficient attention variants, making the advantage harder to calibrate.

Limited analysis of generalization beyond strongly cyclic data; no fallback strategy when cycles are unreliable.

Questions for authors:

Provide ablations for each component: cycle loss vs. sequence loss, mask rate, teacher (EMA), query-attention vs. simple pooling, and patch size.

Add comparisons to recent time-series baselines and to efficient attention models.

Clarify masking (per sequence vs. per cycle), hyperparameters, and data split protocols to avoid leakage.

**Justification:**

MACE tackles a significant bottleneck in long time-series modeling with a straightforward approach that enhances both speed and results for genuine cycles. While the initial evaluation shows promise, it lacks ablations, detailed compute metrics, segmentation robustness, and broader baselines, which weakens the claims. Nonetheless, the core contribution is solid and warrants publication if the authors address these gaps.

---

> ### Author Rebuttal · Authors · 2025-10-21
>
> Thank you very much for taking the time to review our work and providing useful feedback. I respond to each of your points below:
> 1. Cycle detection robustness
>     * This is a good point and we will add a more detailed description on how the cycles are defined in each dataset. We did not find a straightforward way to test the effect of the boundaries in our work, and we believe that a more complex study is required and that this should be explored in future work.
> 2. Measure computational usage
>     * This is a good point, and will be considered in future work, but due to time limitations, the experiments cannot be repeated to measure the computational usage.
> 3. Attention variants
>     * I think low-rank attention variants could be very interesting for future work, as they might be able to eliminate some of the potential bias introduced by patching; however, this falls outside the scope of this work.
> 4. Non-cyclic data
>     * Excellent point, I think if the model could detect when data is not sufficiently cyclic, it would be considerably more useful and  should be taken up in future work.
> 5. Ablation
>     * We will add ablations for cycle loss vs. sequence loss, mask rate, teacher encoder (EMA) and query-attention vs simple pooling. Patching will not be considered in this work as it falls outside the scope, but should be studied further as it is the default in masked autoencoder models in time series.
> 6. Clarifications
>     * We will add some more information on how the masking is performed, on the hyperparameters used and how data splits are performed.

---

### Official Review · Reviewer_FgbH · 2025-09-29
**Lack key ablations**

**Rating:** 4
**Confidence:** 2

**Summary:**

This paper proposes the Masked Autoencoder Cyclic Encoder (MACE), which innovatively takes the cyclic structure of time-series data as an inductive bias. It decomposes sequences into cyclic units and performs masked reconstruction at both the cycle and sequence levels to balance the computational efficiency and performance of long time-series modeling.

**Strengths:**

1. The paper integrates cyclic structure as an inductive bias into masked autoencoders. By splitting long time-series into cyclic units, it effectively reduces the quadratic complexity of Transformers in processing long sequences. Meanwhile, the dual-loss design ensures the accuracy of cycle-level and sequence-level modeling, with validity verified across multiple cyclic data scenarios.

2.  On large cyclic datasets MACE not only outperforms mainstream baselines in performance but also significantly shortens fine-tuning time. This addresses the pain point of high computational costs for long time-series Transformer models, holding practical application value.

**Weaknesses:**

1. The paper does not explore the impact of key designs: fixed 75% sequence masking ratio, cycle segmentation methods  on performance, nor does it verify the necessity of the teacher-student architecture. Thus, the rationality of core designs lacks sufficient experimental support.

2. On small datasets like Air quality, unoptimized computational overhead (from operations such as cycle splitting and padding) makes MACE’s fine-tuning efficiency significantly lower than that of MOMENT (SVM, 19 seconds) and TimeMAE (21 seconds), limiting its use in small-data scenarios.

**Justification:**

Through clear architectural design (cycle decomposition for pre-training + cycle class token utilization for fine-tuning), experimental validation on multiple types of cyclic datasets, and dual improvements in efficiency and performance, the paper demonstrates the value of the "cyclic inductive bias" in time-series masked autoencoders. It responds to the field's demand for efficient long time-series modeling, holding both theoretical and practical significance.

---

> ### Author Rebuttal · Authors · 2025-10-21
>
> Thank you very much for taking the time to review our work and providing useful feedback. I respond to each of your points below:
>
> 1.	We will conduct an ablation experiment to test various masking ratios, as well as evaluate the performance with and without the teacher encoder. We did not find a straightforward way to test the effect of the cycle segmentations in our work, and we believe that a more complex study is required and that this should be explored in future work.
> 2.	Yes, I believe this is caused by how the model is implemented in PyTorch, which necessitates reshaping and padding the sequences of varying lengths, contributing to the computational overhead in part.

---

### Official Review · Reviewer_sqRU · 2025-10-06

**Rating:** 4
**Confidence:** 3

**Summary:**

The paper introduces MACE (Masked Autoencoder Cyclic Encoder), a time-series pretraining method that builds an explicit cycle inductive bias. Long sequences are segmented into cycles, heavily masked, and learned with a teacher–student MAE using two reconstruction targets: (1) masked regions in a cycle-level embedding space and (2) the original sequence space. For fine-tuning, only the per-cycle class tokens are kept and aggregated by a learnable query token for downstream classification, reducing effective sequence length and compute. Across several cyclic datasets (e.g., ECG, energy, air quality), MACE reports consistent F1 improvements and faster fine-tuning; on semi-cyclic data, it is competitive but not always best, and performance depends on the quality of cycle segmentation.

**Strengths:**

1. Treating cycles as first-class units both reduces attention length and regularizes representation learning for inherently periodic data (ECG, load, weekly air quality). The design is well specified.
2. Using only cycle class tokens and a query token for sequence-level prediction is computationally lean and conceptually neat.
3. Consistent F1 improvements on four cyclic datasets and faster fine-tuning on PTB-XL are documented; the study includes dataset descriptions and protocol details (5-fold CV, early stopping).

**Weaknesses:**

1. Datasets use heterogeneous procedures (timestamps, heartbeat detection via NeuroKit, Savitzky–Golay + find_peaks). The method’s sensitivity to boundary noise or mis-segmentation is not quantified via ablations or noise injection.
2. MOMENT is not fully fine-tuned due to cost (linear probe or SVM only), and some probes are stopped/estimated, which may overstate MACE’s advantage; stronger modern efficient baselines (e.g., low-rank attention/SSM variants mentioned in text) are not compared empirically.
3. The focus is F1; confidence intervals or significance tests are absent. Table formatting suggests a possible typo in UCI HAR results (e.g., “90.96±0.”).
4. On semi-cyclic datasets, MACE is competitive but not best—especially vs CRT on FordA when leveraging FFT phase—highlighting that the method’s bias can be mismatched to data without clean periodicity.

**Justification:**

This paper addresses an important gap in time-series pretraining by injecting an explicit cycle inductive bias into a MAE framework and aligning training and fine-tuning to that bias. The idea—segmenting sequences into cycles, learning with dual reconstruction targets (cycle-embedding and original space), and aggregating only cycle class tokens at fine-tuning—is technically sound and well motivated for domains with clear periodicity (ECG, load, air quality). The reported gains on multiple cyclic datasets, coupled with reduced fine-tuning wall-time on long, multi-channel data, indicate practical value and a favorable accuracy–efficiency trade-off in those regimes.

However, the contribution’s impact is constrained by several evaluation gaps. Performance appears sensitive to cycle segmentation, yet robustness to boundary noise is not quantified. Compute claims rely on wall-clock timings without standardized FLOPs/memory or end-to-end cost, limiting generality. Ablations are insufficient to isolate the contributions of key design choices (cycle vs sequence losses, teacher–student, masking ratio, patch size, query-over-class-tokens). Finally, baselines omit strong modern efficient sequence models and provide only partial settings for some methods, which clouds fairness.

Overall, the method is promising for periodic data; its broader significance would be clearer after robustness analyses, fuller ablations, standardized efficiency metrics, significance testing, and stronger baseline comparisons.

---

> ### Author Rebuttal · Authors · 2025-10-21
>
> Thank you very much for taking the time to review our work and providing useful feedback. I respond to each of your points below:
>
> 1. This is a good point and we will add a more detailed description on how the cycles are defined in each dataset. We did not find a straightforward way to test the effect of the boundaries in our work, and we believe that a more complex study is required.
>
> 2. We will add some additional experiments for the running times to provide better estimates and include standard deviations using k-fold. The full fine-tuning of MOMENT was still prohibitively expensive. I think low-rank attention variants could be very interesting for future work, as they might be able to eliminate some of the potential bias introduced by patching; however, this falls outside the scope of this work.
> 3. The typo has been fixed and standard deviations are included in the tables.
> 4. This is part of the inductive bias of the model, but it would be interesting in future work to detect at what point the data is not cyclic “enough” for MACE to give a performance boost.

---

### Official Review · Reviewer_LSKK · 2025-10-09
**MACE, a masked autoencoder tailored to cyclic time-series**

**Rating:** 4
**Confidence:** 4

**Summary:**

This work introduces MACE, a masked autoencoder for time series that embeds a “cycle” inductive bias. From my understanding, the procedure is that the sequences are segmented into cycles, patched with positional embeddings and a per-cycle class token, then trained under high sequence-level masking to reconstruct both (i) teacher-guided cycle embeddings and (ii) the full sequence reconstituted from cycles; the teacher is an EMA of the student, giving stable, unmasked targets while losses are computed only on masked regions, which is a sound MAE practice that limits information leakage from visible inputs.

This two-view objective is explicitly defined and simple (MSE in embedding and data space), and the architecture clearly shortens effective encoder length by operating first per-cycle and then over cycle tokens, which is a coherent way to tie the claimed efficiency to the inductive bias. Fine-tuning reuses the encoder and forms sequence representations by querying over the set of cycle class tokens, matching the prior and avoiding mismatched pretrain/finetune objectives.

Empirically, the method is evaluated on strongly cyclic datasets (e.g., PTB-XL ECG), with the abstract claiming superior accuracy/F1 vs. three MAE-style baselines and reduced training time on large data; these task designs are reasonable for testing the hypothesis that cycles help, but the overall correctness hinges on proper cycle boundary detection and padding, strict subject/household-level splits, and matched masking/optimization across methods.

Given the clear objective, EMA teacher setup, masked-only losses, and an evaluation suite aligned to the stated bias, the contributions are technically sound; the main risks to validity are data segmentation and split leakage rather than flaws in the learning objective itself.

**Strengths:**

This paper’s core strength is a clean, clear objective that tightly matches its stated inductive bias. By decomposing sequences into cycles, masking at the sequence level while allowing the mask to distribute across cycles, and training a student encoder to reconstruct both the teacher’s cycle-space embeddings and the reconstituted full sequence in data space, the method avoids information leakage, provides complementary self-supervised signals, and keeps the math and implementation simple and auditable. The EMA teacher gives stable, unmasked targets, and the loss is precisely defined (MSE over masked indices for both the embedding and signal reconstructions), which supports correctness and reproducibility.

A second strength that I noticed is architectural coherence from pre-training to fine-tuning. It seems that the pre-training treats cycles as first-class units (patched with absolute position encodings and a per-cycle class token), which naturally reduces the effective attention length and focuses representation capacity on local periodic structure; fine-tuning then reuses the same encoder and forms sequence-level representations by attending over the learned cycle tokens, so there is no pretrain/finetune mismatch. This is not just elegant; it is mechanistically aligned with the claimed efficiency gains and with tasks where cycle-level variations carry class signal.

On quality, the paper provides concrete training protocols (5-fold CV, early stopping), and a useful efficiency analysis: fine-tuning time is “considerably lower” on the largest dataset (e.g., PTB-XL) compared to baselines, which aligns with the reduced attention length afforded by cycle processing. These details matter because they connect the proposed bias to real compute savings, not just accuracy.

On significance, the contribution is not “a bigger transformer” but a principled inductive bias that yields both accuracy and speedups on long, cyclic signals. Since the technique is modular and avoids brittle tricks, it is likely to transfer to related domains where periodic components dominate, making the work meaningful beyond the specific datasets tested.

I'm curious to know how exactly cycle boundaries are identified in each dataset, and how robust the results are to boundary errors (e.g., peak-finder thresholds)? I was wondering if masking rates are verified to match at the sequence level across all methods, and are optimizer, augmentation, and compute budgets are matched for fair comparison? And can you provide any ablations isolating each ingredient (e.g., teacher-student vs. single encoder) to quantify where the gains originate? I'm curious to know more about these things!

**Weaknesses:**

From my understanding, the most consequential risk is that several problem formulations and preprocessing choices can silently bias results if not tightly controlled. The UCI Electricity Load task defines labels by the household’s year-level consumption quantile and then classifies week-long segments; unless the train/test split is strictly by household, the target implicitly encodes information aggregated across the entire year that could leak across folds or create an overly easy discrimination signal unrelated to per-week patterns.

Even with per-household splits, the label itself is defined using the full year, so the task departs from a realistic prospective setting and may inflate apparent gains from cycle-aware representations; I'd need explicit confirmation of household-level isolation, exact fold construction, and whether labels are computed using only data on the training side of a split.

Another weakness I noticed is cycle boundary detection and padding. From my understanding, the method relies on accurate cycle start indices and uniform padding before patching and masking; small systematic errors in boundary placement or padding policy can create alignment artifacts that a transformer can exploit during pretraining, thereby overstating the value of the cyclic inductive bias itself. The paper shows that sequences are split into cycles and padded to equal length, then patched with absolute positional embeddings and a prepended cycle class token; this is straightforward, but the empirical sensitivity to boundary noise, padding length, and patch size is not quantified, leaving it unclear whether the gains persist under more realistic, noisy segmenters or when cycle lengths vary more widely.

I'd like to see the authors' comments on the above two points! Also, I'd like to know if the authors could provide diagnostics verifying the realized 75% sequence-level masking (and its distribution across cycles) and show that identical mask policies are used for all baselines.

**Justification:**

My judgment is based on how well the method, experiments, and claims fit together. The idea is clear and technically sound. They're breaking a signal into cycles, masking most of it, and learning from a stable teacher so the model can rebuild both cycle features and the full signal. This matches the kinds of data used (strongly periodic signals like ECG) and explains the reported accuracy gains and faster fine-tuning. However, a few details could still affect the results, such as how cycle boundaries are found and padded, how the data splits and labels are made (to avoid leakage), and whether masking and training settings are truly matched across baselines. Missing ablations also make it hard to pinpoint exactly which parts of the method drive the improvements. Because the core is strong but some checks are not yet shown, I'd judge the paper as promising and likely correct for cyclic data, with acceptance warranted if the authors confirm the data hygiene, add the key ablations, and provide basic efficiency numbers. I'd also kindly ask the authors to maybe answer the questions that I proposed above!

---

> ### Author Rebuttal · Authors · 2025-10-21
>
> Thank you very much for taking the time to review our work and providing such detailed and useful feedback. I have tried to break down your comments and respond to them one by one below:
> 1. Boundary detection robustness and patching
>     * This is a good point and we will add a more detailed description on how the cycles are defined in each dataset. We did not find a straightforward way to test the effect of the boundaries in our work, and we believe that a more complex study is required, which can be explored in further work. We will also add values on the patch sizes in the paper.
> 2. Matching masking rates across models
>     * The masking rates are the same across all compared models. Optimisers are set to the defaults in the code presented by the authors of the other models. I am slightly unsure of what you mean by augmentation here. The accessible resources are the same across all models during the testing of the running time.
> 3. Teacher-student vs single encoder
>     * We will conduct a brief ablation study to evaluate the model without a teacher encoder, testing its impact on performance.
> 4. UCI electricity
>     * The train-test split is done strictly on the household level; we will add a sentence to make this clearer. The task deviates slightly from a realistic scenario as the labels are calculated over the entire set, but creating train-test splits across households in the same time frame is also a slightly strange scenario, as you are then trying to label the test set based on training from a different distribution. Although the task could maybe have been created slightly differently, it should not inflate cycle apparent gains, as all the models are given the same data and should capture the same potential biases.
> 5. Padding
>     * Padding is used in the implementation because PyTorch requires tensors to be of equal length. However, what is passed to the encoder is only the visible part of the sequence, and visible tokens are never selected from the padding. The loss is only calculated over the masked tokens, so there should not be any leakage regarding the varying cycle lengths.
> 6. Masking
>     * We will add a short ablation study testing various masking rates on MACE to test the realised 75% masking. The mask distribution across cycles is the same as if they were one long sequence that was masked and then split into cycles. There are likely some interesting experiments on how the varying of masking rates within cycles affects learning (the masking rate over the entire sequence is still constant), but this should be explored further in future work. I think masking learning dynamics in general is a very interesting area, and I am planning to continue future work in this area.

---

### Meta-Review · Area_Chair_vWRZ · 2025-10-31

**Recommendation:** Accept (Poster)
**Confidence:** 3

**Metareview:**

This paper introduces MACE, a masked autoencoder that integrates cycles as inductive bias. As MAE, MACE consists in a pre-training followed by a fine-tuning. The pre-training architecture combines a student and teacher encoders, the student receiving the masked version of the data fed to the teacher. The data is in this case a sequence split into cycles processed through a "patch, embed, class token" (fig. 3)
Fine-tuning is achieved by using a multi-head attention layer on the class tokens generated by the pre-trained student encoder. The method is tested on several cyclic and semi-cyclic datasets.

Reviewers agree on the efficiency of the inductive bias introduced. Splitting the sequences into cycles alleviates the complexity increase of transformers over long sequences. The results on cyclic datasets are strongly backing the proposed method, both for classification performance and shorter fine-tuning times (especially on long sequences).
There are however a number of concerns raised by the reviewers. In particular the the method's sensitivity to the accuracy of the cycle detection method has to be studied. While the authors acknowledged the issue, they argued that this was no way to test this simply. The influence of the masking ratio is also missing from ablation studies, which should be part of the final version. The paper's results also feel incomplete to most reviewers, as MOMENT has not been fine-tuned (due to processing time constraints), and some other baselines such as low-rank attention are also missing.

Overall, the method shows promising results, and is well devised. Despite the number of concerns (some of which are to be adressed in the final version), it seems the work deserves to be published.

---

### Decision · Program_Chairs · 2025-11-05

**Decision:**

Accept (Spotlight)

**Comment:**

We recommend an oral and a poster presentation given the AC and reviewers recommendations.

A spotlight presentation refers to a poster selected for an oral highlight but not designated as a full oral presentation per the AC’s recommendation.